# Variation of Aroma Components of Pasteurized Yogurt with Different Process Combination before and after Aging by DHS/GC-O-MS

**DOI:** 10.3390/molecules28041975

**Published:** 2023-02-19

**Authors:** Mu Zhao, Hongliang Li, Dongjie Zhang, Jie Li, Rong Wen, Hairan Ma, Tingting Zou, Yaqiong Hou, Huanlu Song

**Affiliations:** 1Laboratory of Molecular Sensory Science, Beijing Technology and Business University, No. 11, Fucheng Road, Haidian District, Beijing 100048, China; 2Inner Mongolia Mengniu Dairy (Group) Co., Ltd., Hohhot 011500, China

**Keywords:** pasteurized yogurt, dynamic headspace sampling (DHS), GC-O-MS, odor-active compounds, r-OAV, PLS-DA

## Abstract

Pasteurized yogurt is a healthy yogurt that can be stored in ambient temperature conditions. Dynamic headspace sampling (DHS) combined with gas chromatography-olfactory mass spectrometry (GC-O-MS), sensory evaluation, electronic nose (E-nose), and partial least squares discriminant analysis (PLS-DA) were used to analyze the flavor changes of pasteurized yogurt with different process combinations before and after aging. The results of odor profiles showed that the sensory descriptors of fermented, sweet, and sour were greatly affected by different process combinations. The results of odor-active compounds and relative odor activity value (r-OAV) showed that the combination of the production process affected the overall odor profile of pasteurized yogurt, which was consistent with the sensory evaluation results. A total of 15 odor-active compounds of 38 volatile compounds were detected in pasteurized yogurt samples. r-OAV results revealed that hexanal, *(E)*-2-octenal, 2-heptanone, and butanoic acid may be important odor-active compounds responsible for off-odor in aged, pasteurized yogurt samples. PLS-DA and variable importance of projection (VIP) results showed that butanoic acid, hexanal, acetoin, decanoic acid, 1-pentanol, 1-nonanal, and hexanoic acid were differential compounds that distinguish pasteurized yogurt before and after aging.

## 1. Introduction

Pasteurized yogurt, also known as room-temperature yogurt, is made by adding a second pasteurization process to the original process of traditional yogurt. O’Brien et al. found no significant differences in immunological parameters and gastrointestinal comfort between pasteurized yogurt and low-temperature yogurt [1]. Additionally, the shelf life of pasteurized yogurt can be as long as 4–6 months under ambient temperature conditions, breaking the limitation of low-temperature yogurt requiring full cold chain transportation and storage and short shelf life. Therefore, pasteurized yogurt has gradually entered the life of consumers in recent years and has become a popular yogurt.

At present, there have been some studies on the effect of the yogurt production process on yogurt. Tian et al. [2] studied the effect of lactic acid bacteria species on the aroma profile of yogurt. Miyaji et al. [3] studied the effect of pasteurization conditions on pasteurized drinking yogurt. Ding et al. [4] studied the characteristics of metabolomics and physicochemical properties of yogurt under aerobic and strict anaerobic fermentation conditions. Yang et al. [5] studied the effect of fermentation temperature on yogurt quality. Xia et al. [6] studied the effect of fermentation time on yogurt quality. Summarizing the above studies, it was found that most of the studies involved fermentation strains and fermentation conditions setting, but there were very few studies on the influence of other processes. Sfakianakis et al. analyzed the odor, flavor, taste, sensory, and volatile components profile of yogurt derived from milk homogenized by ultrasound or pressure, respectively [7]. Therefore, the homogenization process of raw milk will affect the quality of yogurt. In the process of yogurt production, both the homogenization and smooth pump can make the fat particles smaller and the texture smoother. However, the influence of the above two processes on the yogurt flavor has not been reported.

With the presence of large amounts of live bacteria in yogurt, the volatile compounds in yogurt change all the time during storage [8]. Most studies have analyzed the changes of volatile compounds or odor-active compounds during the storage of low-temperature yogurt [9,10,11]. Compared with low-temperature yogurt, pasteurized yogurt has a longer storage time and unlimited storage temperature, so it is more prone to processes such as fat degradation and lipid oxidation during its storage. Milk powder and milk often occur fat oxidation and produce off-odor [12,13]. Although the odor-active off-flavor compounds in aged, pasteurized yogurt have been identified in our previous studies [14], there are still some shortcomings. The previous research results were only discussed and studied based on a group of samples, and it is unclear how the research results would change when the sample range was expanded.

In this study, the aging process was used to simulate the state of pasteurized yogurt before and after storage. In this study, sensory evaluation combined with a variety of instrument analysis was used to comprehensively analyze the flavor characteristics of pasteurized yogurt, and data were further explored by combining with multivariate statistical analysis, so as to realize the purpose of analyzing the changes in aroma components of different process combinations of pasteurized yogurt before and after aging. This study can analyze the effect of the production process on pasteurized yogurt flavor and flavor changes during storage from the perspective of small molecular compounds, which is helpful in improving the production process, prolong shelf life and improve the quality of pasteurized yogurt.

## 2. Results and Discussion

### 2.1. Establishment of Overall Odor Profile

Firstly, the perception of the overall odor of the six pasteurized yogurt samples was evaluated, and a radar chart was drawn (Figure 1). The fermented odor appeared to be the main characteristic odor of the pasteurized yogurt, as evidenced by its higher score by the sensory panelists. Among the samples before aging, bhF sample showed strong characteristics of fermented odor, sour odor, and sweet odor, which have been proven to be beneficial to the overall odor profile of yogurt in previous studies [15,16]. There was no significant difference between creamy odor and milky odor. After aging, all samples showed obvious oxidized and fishy odor and showed a trend of weakening in fermented odor, sweet odor, sour odor, and milky odor compared with the samples before aging, resulting in the decline of the overall odor quality of pasteurized yogurt. There was a similar phenomenon in other dairy products, with the extension of storage time will also appear oxidized odor and other off-odor [12,13]. However, this change does not occur in the later period of low-temperature yogurt storage, which is due to the short shelf life and its strict low-temperature storage conditions, avoiding the production of lipid oxidation and other possible off-odor sources [7]. It was noteworthy that the bhF sample had good flavor before aging but had the strongest oxidized and fishy odor after aging (bhA sample). In summary, pasteurized yogurt samples with different process combinations had different odor profiles and had off-odor after aging. The overall odor profile is formed by the interaction of volatile compounds in the sample, so it is very necessary to study the volatile compounds in pasteurized yogurt samples, which is conducive to clarifying the reasons for the odor changes of pasteurized yogurt from the perspective of small molecular compounds.

### 2.2. Volatiles in Pasteurized Yogurt Analyzed via E-Nose

The electronic nose is equipped with ten different receptors (Table 1), it does not accurately identify or describe a certain odor, only certain compounds, so it is often used to distinguish between different samples. This study was mainly used to investigate differences in volatile components of different pasteurized yogurts. Figure 2A shows the radar chart drawn from the E-nose receptors’ response values. The six samples were similar in shape. However, the ten receptors differed in intensity. Among them, the W5S receptor, which is sensitive to oxynitride, responded most strongly, which may indicate that these compounds were present at high levels and that the W5S receptor was the most important indicator to distinguish the six samples. Specifically, the bhA sample had the highest receptor response value. Figure 2B was the result of PCA, the horizontal axis indicates the contribution rate of PC1 (98.07%), whereas the vertical axis represents the contribution rate of PC2 (1.85%), and the total contribution rate reaches 99.92%. The greater the total contribution rate, the more the data can reflect the original information of the samples [17]. According to the sum of the contribution rates, the two principal components retained the main characteristics and information of the volatiles of the samples, and the contribution of the horizontal axis was larger. The distance between the bhA sample and other samples was large in the PC1 direction, indicating that this sample was quite different from other samples. The other samples showed little difference in the PC1 direction, and there were some differences in the PC2 direction. The volatile components of the three samples, ahF, asF, and bhF, overlapped and intersected, and the volatile components of the three samples were similar to a certain extent. In summary, these results supported the overall odor profile results.

### 2.3. Volatile Compounds in Pasteurized Yogurt Analyzed via GC-MS

According to the content distribution of volatile compounds in six pasteurized yogurt samples, a heat map was drawn, as shown in Figure 3. The contents of 38 volatile compounds in six samples were displayed, and the types and contents of volatile compounds were different, including aldehydes (8), ketones (8), alcohols (7), acids (7), esters (4), and others (4). This result was consistent with other studies [6,15]. Among the three samples before aging, volatile compounds were the most abundant in bhF samples, followed by ahF and asF samples. The contents of acetaldehyde, 2,3-butanedione, 2,3-pentanedione, acetoin, 3-methyl-2-buten-1-ol, etc., in bhF samples were higher than those in ahF and asF samples. These compounds were representative of fine volatile odor compounds in yogurt [15]. The difference in volatile compound content between ahF and asF samples may be caused by the existence of temperature and pressure during homogenization. Specifically, studies have indicated that during milk processing, the degradation of lipids and proteins during hydrostatic pressure treatment can increase the formation of flavor compounds such as aldehydes [18], and the rise of processing temperature can increase the formation of ketones [19]. Therefore, it is possible to explain the higher content of some aldehydes and ketones in the samples produced by the homogenizer (ahF and bhF), which uses pressure and temperature to break up fat particles. After pasteurized yogurt samples were aged, the contents of some volatile compounds showed an obvious increase trend, including aldehydes, ketones, acids, etc. These compounds were proven products of thermal degradation and oxidation of unsaturated fatty acids [8]. It is not difficult to know that a series of reactions, such as fatty acid degradation and lipid oxidation, may occur in the aging process of pasteurized yogurt and ultimately lead to an increase in the content of these compounds. Similarly, the phenomenon has been seen in other foods, such as powdered milk, roasted mutton, bacon, etc. [13,16,20]. Among all the samples, bhA sample had the most abundant volatile compounds, which was consistent with the response value of the E-nose receptor.

### 2.4. Odor-Active Compounds in Pasteurized Yogurt Analyzed via GC-O-MS

The results of the identification of odor-active compounds in all samples are shown in Table 2. Aldehydes are the key carbonyl compound of yogurt, and four aldehyde odor-active compounds were identified, including acetaldehyde, hexanal, *(E)*-2-octenal, and benzaldehyde. In the samples before aging, acetaldehyde (grass-like) and benzaldehyde (almond-like) have higher content. Acetaldehyde is formed in several ways, the most important of which is the breakdown of threonine into acetaldehyde and glycine. As one of the most important odor-active compounds in yogurt, the content of acetaldehyde was significantly different among the three samples of ahF, asF, and bhF. Specifically, bhF was the most abundant sample, followed by ahF and asF. The fermentation process and homogenization process of pasteurized yogurt affect the generation of acetaldehyde. In the aged samples, the content of benzaldehyde and hexanal increased significantly, on the contrary, the content of acetaldehyde did not change significantly. Benzaldehyde may be derived from the degradation of phenylalanine and the oxidation of *α*-linolenic acid [21]. The content of hexanal was even increased by a factor of 100–500. It was formed in milk and yogurt by β-oxidation of the *n*-6 fatty acids linoleic and arachidonic acids [13], which may be facilitated by the aging process leading to a large increase in hexanal content. When the concentration of hexanal is low, it can give food a certain fragrance and fruity odor, but when the concentration reaches 4.5 μg/kg, it shows a grass-like odor, which seriously affects the flavor of the food [22]. Additionally, it was worth noting that *(E)*-2-octenal was newly formed after the aging of the samples. *(E)*-2-Octenal was metabolized by lipoxygenase using linoleic acid as a precursor, which was one of the products of lipid oxidation during aging.

Ketones are mainly formed by thermal degradation of amino acids and oxidation of unsaturated fatty acids. A total of five odor-active compounds of ketones were identified, including 2,3-butanedione, 2,3-pentanedione, 2-heptanone, acetoin, and 2-nonanone. They were buttery, creamy, and sweet, and could provide a positive odor characteristic for pasteurized yogurt. They were formed as follows: 2,3-butanedione and 2,3-pentanedione were produced by the chemical decarboxylation of their precursors, 2-acetolactate and 2-aceto-2-hydroxybutyrate, respectively. Acetoin was readily converted from 2,3-butanedione by diacetyl reductase [9,23,24]. This formation pathway could explain the phenomenon that the content of acetoin was the most abundant and the content of 2,3-butanedione was lower in this study. Acetoin, 2,3-butanedione and 2,3-pentanedione, three of the positive odor compounds recognized as important in yogurt, all showed the highest levels in the bhF samples. Combined with previous studies, it can be known that the heating process of homogenizer promoted the formation of ketones in raw milk [19], kept the promoting effect, and finally made the bhF sample has a high content of ketones. Similarly, Reis et al. also demonstrated that temperature and pressure can significantly affect the formation of ketones in milk [25]. However, the detailed influence process is still unknown, which requires further in-depth research. Such research is very beneficial to improve the quality of yogurt from the production equipment itself. After aging, the contents of acetoin and 2,3-butanedione tended to increase, and the content of 2-heptanone increased significantly. The variation trend of acetoin and 2,3-butanedione was consistent with the results of previous studies [10]. 2-Heptanone is usually formed by β-oxidation of free fatty acids or β-ketoacid decarboxylation, and aging may promote this process and lead to a large increase in its content. In previous studies, a significant increase in 2-heptanone content in other dairy products, such as milk, was found during storage [12].

Acids are essential compounds in yogurt, which can enrich the sour odor of yogurt. In this study, a total of 5 acid compounds showed odorant activity, including acetic acid, butanoic acid, pentanoic acid, hexanoic acid, and octanoic acid. Their odor characteristics were not exactly the same, they could present vinegar-like, sweaty, and cheesy odor. Different scholars hold different opinions on the formation of acids in yogurt. Some researchers believe that C_2_–C_4_ is produced by lactic acid bacteria and lactic acid metabolism, C_4_–C_20_ is mainly formed by fat decomposition, and some researchers believe that the key precursors of most volatile fatty acids are amino acids [15]. Hexanoic acid and octanoic acid were the most abundant acids in pasteurized yogurt samples, which was consistent with the results of Moineau-Jean et al. [10] and Tian et al. [2] but inconsistent with those of Rychlik et al. [11]. It was noted that the content of butanoic acid increased significantly after aging, and new pentanoic acid was produced in ahA sample. It may be caused by the oxidative decomposition of fatty acids in the aging process of pasteurized yogurt, and the higher content of butanoic acid can bring cheese-like off-odor to yogurt.

Styrene (plastic-like) was detected in the samples, and its content increased significantly after aging. It was speculated that styrene may have migrated from the packaging material to the pasteurized yogurt during aging [26]. Moreover, alcohols were also important volatile compounds in pasteurized yogurt (Figure 3), but they have not been identified as odor-active compounds, possibly because they have a higher threshold than other aldehydes and ketones [27] and therefore cannot be detected on the olfactory detection port.

### 2.5. The Contribution Degree of Odor-Active Compounds to the Overall Odor Profile of Pasteurized Yogurt Evaluated by r-OAV

The compounds with r-OAV > 1 are generally considered to be of greater importance to the overall odor profile of the sample [28]. Specifically, the larger the value, the greater the contribution to the overall odor profile of the sample. The r-OAV calculation results are summarized in Table 3. Of all odor-active compounds of samples ahF, asF, and bhF, 2,3-butanedione had the highest r-OAV of 408, 547, and 647, respectively, attributing to the buttery odor of fresh pasteurized yogurt samples. This result was consistent with the previous study results [29]. Secondly, 2,3-pentanedione had a higher r-OAV of between 106 to 142, and its odor attribute was also buttery. Both 2,3-butanedione and 2,3-pentanedione had the largest r-OAV in bhF sample, which may have led to the higher score in the evaluation of overall odor profile (Figure 1). The other compounds, such as hexanal, acetaldehyde, 2-heptanone, and 2-nonanone had the slightly larger r-OAV, which was attributed to the grass-like and sweet odor in fresh pasteurized yogurt samples. Acetoin was the most abundant compound in samples, but its r-OAV was only 10, which was due to its high threshold, so its contribution to the formation of the overall odor profile of fresh pasteurized yogurt was relatively weak.

As shown in Table 3, after pasteurized yogurt aging, there were some odor-active compounds with significantly increased r-OAV. Among all compounds, the most significant change was hexanal, which replaced 2,3-butanedione to become the odor-active compound with the largest r-OAV in the aged, pasteurized yogurt samples. Combining with the production pathway of hexanal, it was speculated that this change was the reason for the oxidized odor of aged, pasteurized yogurt samples. Previous studies have shown that aldehydes were the main contributing compounds to the oxidative flavor of UHT milk [30]. Secondly, the r-OAV values of *(E)*-2-octenal, 2-heptanone, and butanoic acid also increased significantly. *(E)*-2-Octenal had a great influence on the overall odor profile of aged, pasteurized yogurt samples because of its low threshold. The results also confirmed that enals produced by fat oxidation have a great influence on food flavor [31]. Hexanal, *(E)*-2-octenal, 2-heptanone, and butanoic acid all had the highest r-OAV values in the bhA sample, which may have resulted in the strongest oxidized odor of the sample (Figure 1). Overall, the changes in these compounds resulted in changes in the overall odor profile of pasteurized yogurt samples after aging.

### 2.6. Differential Compounds of Pasteurized Yogurt before and after Aging Identified by PLS-DA

PLS-DA is a supervised pattern recognition method, which emphasizes the differences between groups to minimize the differences within groups and better grasp the overall characteristics and change rules of multidimensional data [32]. As shown in Figure 4A,B, a clear separation of pasteurized yogurt samples before and after aging could be observed through a dependable PLS-DA model. The PLS-DA model was established based on the semi-quantitative results of volatile compounds in all samples. The parameters R^2^ and Q^2^ represent the explanatory and predictive abilities of the model, respectively, and the values of R^2^ and Q^2^ should be greater than 0.5. The results are more accurate as R^2^ and Q^2^ approach 1 [33]. The R^2^Y of the PLS-DA model was 0.98, and Q2 was 0.88, which indicated satisfactory explanatory and predictive effects of the PLS-DA model for the classification of pasteurized yogurt samples before and after aging.

Based on PLS-DA model, the variable importance of projection (VIP) diagram of the model was obtained (Figure 4C). The VIP is the vector, used to summarize the total importance of the variable in explaining the model [32]. Specifically, VIP value > 1 will contribute to the model. The abscissa of the VIP result graph in Figure 4C was the CAS number of the compound. The main compounds which contribute to the distinction between the two groups (compounds with VIP value > 1) were butanoic acid, hexanal, acetoin, decanoic acid, 1-pentanol, 1-nonanal and hexanoic acid (arranged in descending order of VIP value). In the previous analysis, butanoic acid and hexanal were found to play an important role in the formation of oxidized odor of aged, pasteurized yogurt samples. Fatty acids with more than four carbons are readily produced by fat degradation [15]. The change of 1-pentanol was consistent with the results of previous studies and was considered to be a product of lipid decomposition [34].

## 3. Materials and Methods

### 3.1. Samples

The pasteurized yogurt sample used in this study was a factory pilot sample produced and provided by Inner Mongolia Mengniu Dairy (Group) Co., Ltd. (Inner Mongolia, China), which could not be purchased in the market, and no other volatile substances were added to the sample. The samples were all obtained in 2022, and the information is listed in Table 4. The basic index information of the sample is as follows: Water content of 78%, protein content of 3.2%, fat content of 3.5%, acidity of 75 °T. The homogenization temperature is 50–70 °C and the total pressure is 1 × 10^7^ –1.8 × 10^7^ Pa. Smooth pump speeds are 20–50 Hz. Aged pasteurized yogurt was obtained by fresh pasteurized yogurt accelerated aging for 13 d. This process is completed by Inner Mongolia Mengniu Dairy (Group) Co., Ltd., and it has been proved that this process can effectively simulate and reproduce the status of pasteurized yogurt after the shelf life when it is normally sold and stored. The specific operation and evaluation criteria are still in the confidential stage, belonging to the category of trade secrets. Samples canned and sealed directly after production are kept in a clean, odor-free refrigerator dedicated to the laboratory. All samples were analyzed within 10 d after production. There are six kinds of samples, 10 copies of each sample, 180 g each, a total of 10.8 kg.

### 3.2. Reagents and Chemicals

Sodium chloride (NaCl, analytical reagent, purity ≥99.5%) was obtained from Sinopharm Chemical Reagent Co., Ltd. (Beijing, China). *n*-Alkanes standard solution (C_7_–C_30_) and 2-methyl-3-heptanone (99% purity) were obtained from Sigma-Aldrich (St. Louis, MO, USA). *n*-Hexane (purity >99%) was obtained from Thermo Fisher Scientific (Waltham, MA, USA).

### 3.3. Establishment of Odor Profiles

The odor profile was adjusted on the basis of previous experimental methods [35]. All members of the sensory panel are from the Molecular Sensory Science Laboratory, Beijing Technology and Business University (Beijing, China), and have received more than 1 year of sensory skills training. There were 12 members in the panel, including 6 males and 6 females, with an average age of 30. The sensory evaluation room is clean, odorless, noise-free, well-lit, 25 °C, 65% relative humidity, and compartments to ensure an independent assessment process for each team member. The members of the sensory panel discussed and agreed that the following 7 sensory descriptors should be used as odor characteristics of the sample and applied to establish odor profiles. The sensory descriptors and the references agreed upon by the panelists are as follows: “Fermented” refers to fresh cheese, “sweet” reference to diluted 2-heptanone standard, “sour” refers to 0.08% citric acid solution, “milky” refers to fresh whole milk, “creamy” refers to whipped cream, “fishy” refers to the fishy smell of raw milk, “oxidized” refers to the diluted butanoic acid standard. A seven-point scale (0–6, with a difference of 1) was used to quantify the odor characteristics of the sample. The 10 g sample that has been standing at 40 ℃ for 20 min is presented to the panel members for evaluation, and the evaluation and score are made on the record sheet. The order of sample submission is random, and each sample is guaranteed to be submitted 3 times.

### 3.4. Volatiles Analysis by Electronic Nose (E-Nose)

A portable electronic nose system PEN3 (Airsense Analytics GmbH., Schwerin, Germany) was used to analyze the volatiles in the samples. Ten receptors with different properties were installed in the electronic nose system and described in Table 1. A 10 g sample of pasteurized yogurt was weighed for the analysis and equilibrated at 40 °C for 20 min before the electronic nose test. The parameters of the electronic nose were set as follows: Cleaning time of 60 s before receptors were tested, preparation time of 5 s, receptor detection time of 180 s, and gas flow rate set to 400 mL/min. Each sample was tested 5 times, and an empty bottle was tested as an empty needle between testing different samples.

### 3.5. Extraction of Volatile Compounds by Dynamic Headspace Sampling (DHS)

Pasteurized yogurt (50 g) and sodium chloride (7.5 g) were mixed in a dynamic headspace vial, followed by the addition of 5 μL of 2-methyl-3-heptanone (0.816 μg/μL) as an internal standard and then quickly sealed. Place on a magnetic stirrer (Thermo Scientific, USA) at 600 rpm while incubating in a constant circulating water bath at 40 °C for 20 min. Then, the upper space of the vial was purged with nitrogen (99.999% purity) at a flow rate of 150 mL/min for 40 min to allow volatiles to adsorb into the Tenax TA tube inserted into the vial. After purging, the Tenax TA tube was removed and purged with nitrogen to remove water. Finally, the Tenax TA tubes were placed in a thermal desorption unit (TDU) (Gerstel, Germany) for analysis. Each sample was analyzed in triplicate.

### 3.6. Gas Chromatography–Mass Spectrometry (GC-MS)/Gas Chromatography–Olfactometry (GC-O) Analysis

GC-MS analysis was performed by GC-MS Model 7890A-7000 (Agilent Technologies, Inc., Santa Clara, CA, USA). Two capillary columns of different polarity were installed for the separation of volatile compounds in GC section, respectively, polar capillary column DB-WAX (30 m × 0.25 mm × 0.25 μm; J&W Scientific, Folsom, CA, USA) and medium-polarity DB-5 capillary column (30 m × 0.25 mm × 0.25 μm; J&W Scientific, Folsom, CA, USA). The carrier gas has a constant flow rate of 1 mL/min and uses ultra-high purity helium (99.999%, Beijing AP BAIF Gases Industry Co., Ltd., Beijing, China) as the carrier gas. The initial temperature of the column box was maintained at 40 °C for 5 min and then increased to 230 °C at the rate of 4 °C/min for 5 min. The back GC injector was set to “splitless” mode. The mass spectrometry source temperature was 280 °C. The electron collision mass spectrometry was generated at 70 ev ionization energy, and the scanning range was 33~350 *m*/*z*.

The olfactory detection port (ODP4, Gerstel, Germany) was connected to the GC section for GC-O analysis. During the whole process of GC-O analysis, three members of the sensory panel (1 male and 2 females) sniffed at the olfactory detection port and recorded the retention time and odor characteristics of the odor in real time.

### 3.7. Identification of Volatile Compounds and Odor-Active Compounds

The target compounds were identified by mass spectrometry retrieval (MS), retention index comparison (RI), and sniffing odor description (O). Mass spectrometry retrieval refers to the retrieval of MS results of target compounds in the NIST17 mass spectrometry database, according to the MS matching degree (>800) and MS structure information to identify the compounds. Retention index comparison refers to comparing the actual RI with the standard RI of the target compound. If the difference is less than 50, it is considered to pass the identification. The actual RI value is calculated by the retention time between the target compound and a series of n-alkanes (C_7_–C_30_) [36,37]. Compounds identified only by MS and RI were identified as volatile compounds and were identified as odor-active compounds when they could be detected at the olfactory detection port.

### 3.8. Quantitation of Compounds

A semi-quantitative method was used to study the changes in volatile compound concentration before and after aging different pasteurized yogurt samples. The concentration of each compound was calculated as follows [38,39]:$$C_{i}=\frac{C_{is}\times A_{j}}{A_{is}}$$
where *C_i_* is the concentration of the compound; *C_is_* is the internal standard concentration of 0.816 μg/μL; *A_j_* is the chromatographic peak area of the compound, and *A_is_* is the chromatographic peak area of the internal standard.

### 3.9. Calculation of Relative Odor Activity Value (r-OAV)

Relative odor activity value (r-OAV) was calculated as the ratio of the relative concentration of each compound to its respective odor threshold (OT) [29,39]. The threshold of the compound used in this study is the threshold in water [27].

### 3.10. Statistical Analysis

The experimental data were analyzed by using Microsoft Excel 2019 (Microsoft Corp., Redmond, WA, USA) in triplicate and expressed as mean ± standard deviation. One-way analysis of variance (ANOVA) and Duncan’s multiple range tests were performed using the IBM SPSS Statistics 26 (IBM., Armonk, NY, USA) to analyze the differences between samples. A *p* ≤ 0.05 was considered statistically significant. The heat map and radar charts were made by Origin 2019 (Origin Lab Inc, Northampton, MA, USA). The result of E-nose of principal component analysis (PCA) was generated using the software Winmuster (version 1.6.2). The partial least squares discriminant analysis (PLS-DA) was performed by SIMCA 14.1 (MKS Instruments AB, MA, USA).

## 4. Conclusions

In this study, dynamic headspace sampling (DHS) combined with gas chromatography-olfactometry-mass spectrometry (GC-O-MS) were employed to analyze the flavor changes of pasteurized yogurt with different process combinations before and after aging. A total of 15 odor-active compounds of 38 volatile compounds were detected in six pasteurized yogurt samples. Sensory evaluation and GC-O-MS results showed that under the same fermentation process, the overall odor profile of pasteurized yogurt samples obtained by a smooth pump was better than that obtained by a homogenizer. r-OAV results revealed that hexanal, *(E)*-2-octenal, 2-heptanone, and butanoic acid may be important odor-active compounds responsible for off-odor in aged, pasteurized yogurt samples. PLS-DA and VIP results showed that butanoic acid, hexanal, acetoin, decanoic acid, 1-pentanol, 1-nonanal, and hexanoic acid were differential compounds that distinguish pasteurized yogurt before and after aging. As this study reported the effect of the production process on the overall odor profile of pasteurized yogurt and the variation of odor-active compounds before and after aging, it may provide some useful information for extending shelf life and improving the quality of pasteurized yogurt. Differential odor compounds can be used as odor markers to evaluate the odor quality of pasteurized yogurt and can be used as an evaluation basis to optimize the production process.

## Figures and Tables

**Figure 1 molecules-28-01975-f001:**
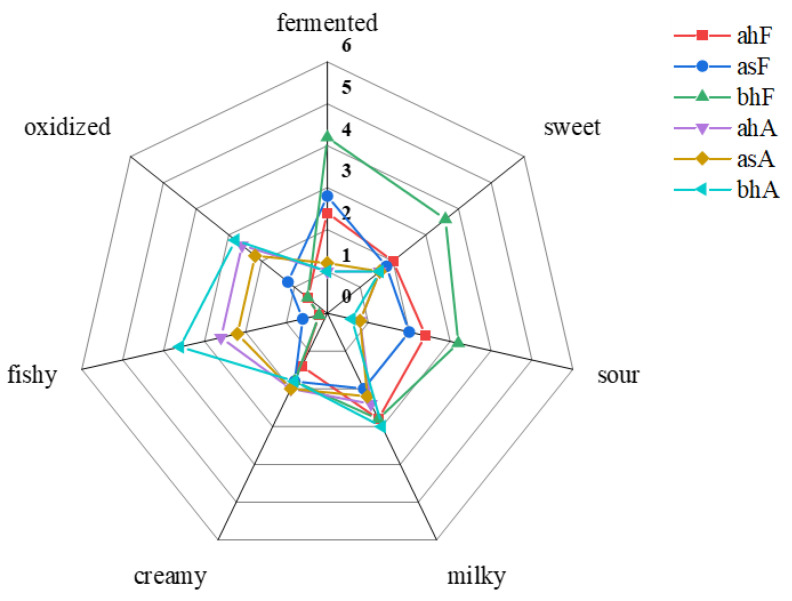
Odor profile evaluation of the six samples.

**Figure 2 molecules-28-01975-f002:**
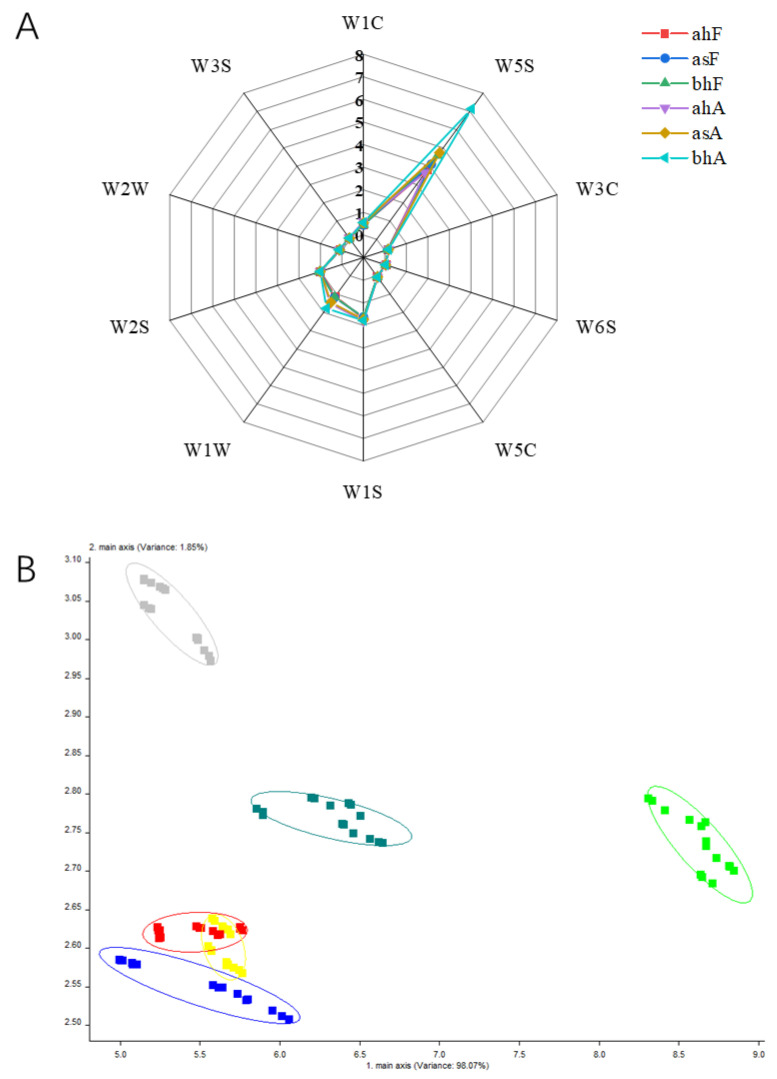
Radar chart and PCA analysis by E-nose. (**A**) Radar chart of the E-nose response of different types of volatiles for different pasteurized yogurt samples, (**B**) PCA plot of the E-nose for different pasteurized yogurt samples.

**Figure 3 molecules-28-01975-f003:**
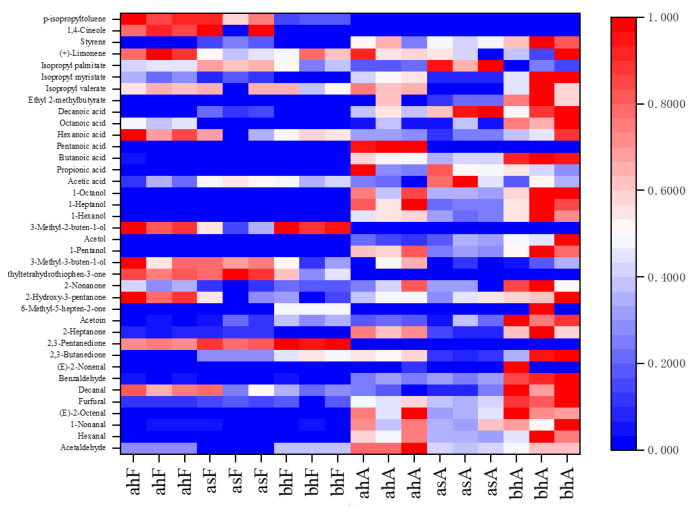
The heat map analysis of the six samples. Every square represents a volatile compound. If a compound had a lower concentration, the color of the square is close to blue, otherwise, the color is close to red.

**Figure 4 molecules-28-01975-f004:**
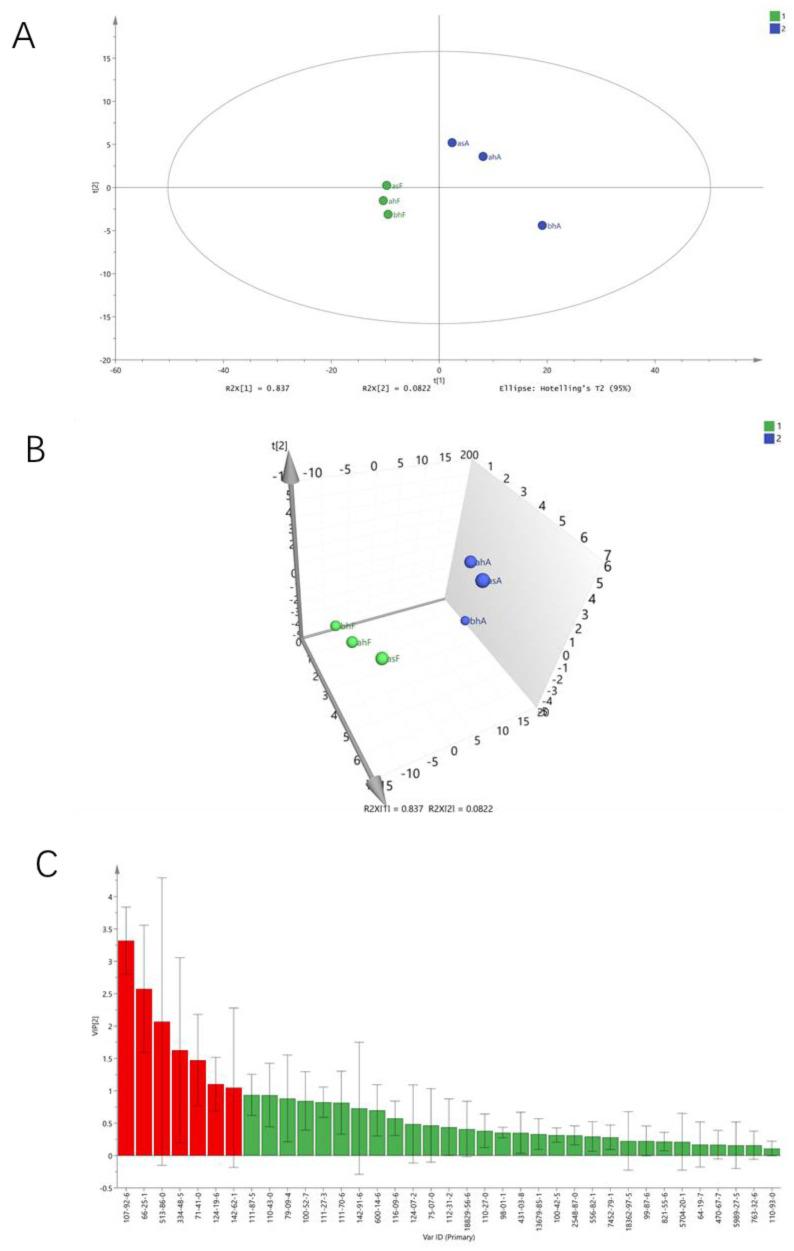
2D (**A**) and 3D (**B**) score charts of PLS-DA and VIP diagram (**C**; red means VIP > 1, green means VIP < 1) of the six samples (R^2^X = 0.919, R^2^Y = 0.98, and Q^2^ = 0.88).

**Table 1 molecules-28-01975-t001:** The description of receptor on E-nose.

No.	Receptor	Specifification
R1	W1C	aromatic hydrocarbon
R2	W5S	broad range
R3	W3C	aromatic ammonia
R4	W6S	hydrogen
R5	W5C	arom-aliph
R6	W1S	broad-methane
R7	W1W	sulfur-organic
R8	W2S	broad-alcohol
R9	W2W	sulfur-chlor
R10	W3S	methane-aliph

**Table 2 molecules-28-01975-t002:** Odor-active compounds determined in six pasteurized yogurt samples by GC-O-MS.

No.	CAS	Compounds	Odor	Identification	RI ^a^		Concentration (μg/g) ^b^					
					DB-WAX	DB-5	ahF	asF	bhF	ahA	asA	bhA
1	75-07-0	Acetaldehyde	grass-like	MS/RI/O	<800	-	1.71 ± 0.00145 ^b^	0.545 ± 0.0237 ^c^	2.16 ± 0.0357 ^a^	4.13 ± 0.469 ^a^	2.28 ± 0.0834 ^a^	2.99 ± 0.264 ^a^
2	431-03-8	2,3-Butanedione	pungent buttery	MS/RI/O	980	<800	2.45 ± 0.0105 ^b^	3.28 ± 0.0248 ^a^	3.88 ± 0.136 ^a^	4.03 ± 0.117 ^a^	2.75 ± 0.0350 ^b^	4.63 ± 0.861 ^a^
3	600-14-6	2,3-Pentanedione	buttery	MS/RI/O	1053	<800	2.12 ± 0.0507 ^a^	2.43 ± 0.116 ^a^	2.84 ± 0.0757 ^a^	-	-	-
4	66-25-1	Hexanal	grass-like	MS/RI/O	1079	803	0.423 ± 0.0427 ^c^	0.124 ± 0.0408 ^c^	0.125 ± 0.00338 ^c^	53.6 ± 9.62 ^ab^	30.3 ± 1.32 ^b^	64.5 ± 19.8 ^a^
5	110-43-0	2-Heptanone	sweet	MS/RI/O	1179	888	2.56 ± 0.350 ^b^	2.94 ± 0.391 ^b^	1.43 ± 0.189 ^b^	13.1 ± 0.990 ^a^	3.18 ± 0.631 ^b^	13.8 ± 3.10 ^a^
6	100-42-5	Styrene	plastic-like	MS/RI/O	1246	-	0.184 ± 0.00218 ^c^	0.475 ± 0.0538 ^b^	0.180 ± 0.00937 ^c^	0.892 ± 0.247 ^a^	0.880 ± 0.0618 ^a^	1.44 ± 0.251 ^a^
7	513-86-0	Acetoin	mild creamy	MS/RI/O	1281	<800	68.1 ± 1.53 ^c^	76.7 ± 7.03 ^bc^	93.7 ± 4.60 ^b^	82.4 ± 2.03 ^b^	84.6 ± 11.1 ^b^	138 ± 8.23 ^a^
8	821-55-6	2-Nonanone	sweet	MS/RI/O	1381	-	0.443 ± 0.0730 ^b^	0.156 ± 0.00409 ^c^	0.235 ± 0.0148 ^bc^	0.569 ± 0.194 ^ab^	0.384 ± 0.00406 ^b^	0.913 ± 0.236 ^a^
9	2548-87-0	*(E)*-2-Octenal	fresh	MS/RI/O	1420	-	-	-	-	0.834 ± 0.246 ^ab^	0.423 ± 0.0638 ^b^	0.911 ± 0.174 ^a^
10	64-19-7	Acetic acid	vinegar-like	MS/RI/O	1461	<800	1.35 ± 0.194 ^a^	1.79 ± 0.0537 ^a^	1.63 ± 0.0837 ^a^	1.23 ± 0.163 ^a^	2.12 ± 0.331 ^a^	1.54 ± 0.213 ^a^
11	100-52-7	Benzaldehyde	almond-like	MS/RI/O	1509	956	1.65 ± 0.248 ^c^	1.49 ± 0.0684 ^c^	1.41 ± 0.213 ^c^	4.58 ± 0.280 ^b^	4.83 ± 0.508 ^b^	12.3 ± 0.830 ^a^
12	107-92-6	Butyric acid	sweaty	MS/RI/O	1630	<800	8.05 ± 0.772 ^c^	5.54 ± 1.38 ^cd^	3.63 ± 0.157 ^d^	80.0 ± 4.75 ^b^	59.7 ± 3.96 ^b^	143 ± 4.18 ^a^
13	109-52-4	Pentanoic acid	sweaty	MS/RI/O	1734	912	-	-	-	1.42 ± 0.0243	-	-
14	142-62-1	Hexanoic acid	cheesy	MS/RI/O	1851	984	17.9 ± 2.58 ^a^	10.3 ± 5.44 ^ab^	13.4 ± 0.526 ^a^	9.49 ± 0.327 ^ab^	8.11 ± 0.966 ^b^	13.9 ± 3.36 ^a^
15	124-07-2	Octanoic acid	cheesy	MS/RI/O	2083	1174	11.8 ± 0.255 ^a^	9.59 ± 0.0197 ^b^	9.51 ± 0.00387 ^b^	10.3 ± 0.824 ^ab^	7.69 ± 0.371 ^b^	13.7 ± 0.763 ^a^

^a^ The retention index (RI) on capillaries DB-WAX and DB-5. ^b^ Mean values of triplicates with standard deviations. Different lower letters in the same row (such as a,b and c) indicate significant differences (*p* ≤ 0.05).

**Table 3 molecules-28-01975-t003:** Odor thresholds (OT) and relative odor activity values (r-OAV) of odor-active compounds in six pasteurized yogurt samples.

CAS	Compounds	OT (mg/kg)	r-OAV					
			ahF	asF	bhF	ahA	asA	bhA
431-03-8	2,3-Butanedione	0.006	408	547	647	672	458	772
600-14-6	2,3-Pentanedione	0.02	106	122	142	-	-	-
66-25-1	Hexanal	0.005	84	24	24	10,726	6062	12,896
75-07-0	Acetaldehyde	0.063	27	9	34	66	36	47
110-43-0	2-Heptanone	0.14	18	21	10	93	23	99
821-55-6	2-Nonanone	0.041	10	4	6	14	9	22
513-86-0	Acetoin	8	9	10	12	10	11	17
100-52-7	Benzaldehyde	0.35	5	4	4	13	14	35
124-07-2	Octanoic acid	3	4	3	3	3	3	5
100-42-5	Styrene	0.065	3	7	3	14	14	22
107-92-6	Butanoic acid	2.4	3	2	2	33	25	60
142-62-1	Hexanoic acid	18	1	1	1	1	<1	1
64-19-7	Acetic acid	22	<1	<1	<1	<1	<1	<1
2548-87-0	*(E)*-2-Octenal	0.003	-	-	-	277	140	303
109-52-4	Pentanoic acid	1.207	-	-	-	1	-	-

**Table 4 molecules-28-01975-t004:** Detailed information of six pasteurized yogurt samples.

Sample Type	Process Combination	Abbreviation
Fresh pasteurized yogurt (F)	Fermentation process a & homogenizer treated (ah)	ahF
	Fermentation process a & smooth pump treated (as)	asF
	Fermentation process b & homogenizer treated (bh)	bhF
Aged pasteurized yogurt (A)	Fermentation process a & homogenizer treated (ah)	ahA
	Fermentation process a & smooth pump treated (as)	asA
	Fermentation process b & homogenizer treated (bh)	bhA

## Data Availability

The authors will make the raw data supporting the conclusions of this manuscript available to any qualified researcher.

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
