# Peer review of "Variation of Aroma Components of Pasteurized Yogurt with Different Process Combination before and after Aging by DHS/GC-O-MS"

_molecules, 2023, doi:10.3390/molecules28041975_

Round 1

Reviewer 1 Report

The manuscript titled “Variation of aroma components of pasteurized yogurt with different process combination before and after aging by DHS/GC-O-MS- by Zhao et al” deals with the effect of production process on pasteurized yogurt flavor and flavor changes during storage from the perspective of small molecular compounds, which is helpful to improve the production process, prolong shelf life and improve the quality of pasteurized yogurt. The subject is interesting and presents some novelty. However, there are some weak points in the research, concerning overall design of the experiment and analytical chemistry. The article was not found suitable for publication in Molecules in the present form, it needs major revisions.

My remarks about the text are as follows:

Line 106: I cannot see the figures in the paper and suppl. material.

Line 168: Adda refrence for:  Benzaldehyde may be 168 derived from the degradation of phenylalanine and the oxidation of α-linolenic acid (Zannou et al., 2020).

Zannou, Oscar, Hasim Kelebek, and Serkan Selli. "Elucidation of key odorants in Beninese Roselle (Hibiscus sabdariffa L.) infusions prepared by hot and cold brewing." Food research international 133 (2020): 109133.

Line 286: Add the obtaining year of the pasteurized yogurt sample.

Line 291: How did the researchers decide on the pasteurized yogurt accelerated aging process in the study? Please explain the criteria.

Line 293: Add the total amount of the samples.

Line 303: Describe members of the sensory panel? Total numbers of panelists, average age and sex.

Line 311: Provide the sensory room properties.

Line 326: The concentration of volatile compounds shown in Tables are by reference to a single internal standard (2-methyl-3-heptanone (0.816 μg/μL)). Is it enough only one internal standard for all compounds? What is the recovery yield of the internal standard? Please explain.

Line 350: Add the sniffer number.

Author Response

Dear Reviewer

The manuscript is revised according to your comments, please have a check. If more are needed, please let me know.

Sincerely Yours

Huanlu Song

Reviewer 2 Report

This MS (molecules-2203268) evaluated the effect of process combination on the aromas of pasteurized yogurt using the GC-O-MS, sensory evaluation and E-nose. These findings would be useful to improve the sensory profile of pasteurized yogurt. The organization and writing of this MS are adequate. Yet some points need to be addressed before consideration for acceptance.

1.      Abstract, Line 14-Line 16, DHS could be used with a combination of sensory evaluation, E-nose or PLS-DA? These sentences were confusing. Please reorganize these sentences.

2.      Line 18-line 19, what is the selection? Line 19, please clarify the sensory profile.

3.      Line 23, what is the VIP? Please provide the full name. Authors are required to detail any abbreviations at the first time that they appear in the text. Remember that readers not always are familiar with the terminology used. Please check this MS.

4.      Introduction, you should clearly state the novelty of this work. From Line 39 to Line 52, you provided previous studies about the changes of yogurt volatiles during the production. However, you did not give clear indications about the necessity of the homogenization and smooth pump. In this section, you should tell the readers that why you conducted this work, not little reported studies about the identification of aroma-active compounds!

5.      Line 63-Line 72, Not adequate the procedures in objective. Something would be better in Materials and methods. Please rewritten this, just specify what are you going to do.

6.      The major concern of this work is lack of the results of Figure.

7.      Results and discussion, results you need just give the results, not to repeat your methods again here. Otherwise, it means that you did not write your methods clearly enough.

8.      You should give more indications about the effect of process combination not the aging and storage in section 2.3 and section 2.4. Recently published related papers should be discussed in the manuscript.

9.      Line 135-Line 137, please give more discussions about the effect of temperature and pressure.

10.   Section 2.6, PLS-DA was used to analyze the results of rOAV?

11.   The figure of experimental design should be provided. In this way, readers can quickly understand the main content of the study.

12.   Section 3.1, these samples could be purchased from the market? how long these samples were storage before the analysis of volatiles? Were other volatiles added into these commercial samples? Did they have a long storage life? What is the difference of fermentation process a and b

13.   Section 3.5-3.6, how were these experiments conditions set? Were both of them defined based on preliminary tests, or based on the literature?

14.   Section 3.9, why you calculate relative odor activity value? OAVs were widely accepted by the researchers not the rOAV. Please give more indications.

15.   Line 383, p should be Lower case.

16.   Line 388, it seemed to be a summary not a conclusion.

Author Response

Dear Reviewer

The manuscript is revised according to your sugesstions. If moe are needed, please let me know.

Reviewer 3 Report

The authors of the manuscript have described the results of their research work, which may have application potential. The presented methodology of the work and the discussion of the obtained data, as well as their documentation, are interesting for a wide audience. As far as my comments are concerned, it seems to me that the authors could have emphasized the elements of novelty in the presented manuscript and can give a comment to their earlier publication Characterization of key odor-active off-flavor compounds in aged pasteurized yogurt by sensory-directed flavor analysis. J Agric Food Chem. 2022 Nov 16;70(45):14439-14447. doi: 10.1021/acs.jafc.2c03409.

Author Response

Dear Reviewer

The manuscript is revised according to your comments, please have a check. If more are needed, please let me know.

Round 2

Reviewer 1 Report

I reviewed the article titled “Variation of aroma components of pasteurized yogurt with different process combination before and after aging by DHS/GC-O-MS" again as a referee. The researchers have made necessary corrections.Therefore, I can recommend the manuscript for publication in the Molecules.

Reviewer 2 Report

This MS can be accepted